# Revealing potential interfering genes between abdominal aortic aneurysm and periodontitis through machine learning and bioinformatics analysis

Zhifeng Wu[1,2‡], Fan Zhang[2‡], Yuan Wang[3‡], Chunjiang Liu[2,4], Zhaokun Sun[2], Xiaoqi Tang[2], Liming Tang [ID][2,4]*

1 Department of Hepatobiliary Surgery, Shaoxing Second Hospital, Shaoxing, Zhejiang, China, 2 School of Medicine, Shaoxing University, Shaoxing, Zhejiang, China, 3 Department of intervention Vascular, Hefei Hospital Affiliated to Anhui Medical University, Hefei, China, 4 Department of General Surgery, Division of Vascular Surgery, Shaoxing People's Hospital, Shaoxing, China

‡ These authors contributed to the work equllly and should be regarded as co-first authors.
* tanglm@usx.edu.cn

## Abstract

This study aimed to identify potential interacting genes between abdominal aortic aneurysm (AAA) and periodontitis. To achieve this, we obtained datasets of AAA and periodontitis from the GEO database, conducted differential analysis on the AAA dataset, and performed weighted gene co-expression network analysis (WGCNA) on the periodontitis dataset to preliminarily identify interacting genes via intersection. Subsequently, we refined key candidate genes by constructing a PPI network and applying three machine learning algorithms. These candidate genes were further validated through external independent datasets, receiver operating characteristic (ROC) curves, and Nomograms. Finally, single-gene Gene Set Enrichment Analysis (GSEA), immune landscape analysis, and targeted drug prediction were performed on the identified key genes. In our study, a total of 323 differentially expressed genes (DEGs) related to AAA and 4,412 periodontitis-related module genes were identified, producing 90 interacting genes through intersection initially. Through PPI network analysis and machine learning, we prioritized 7 key interacting genes. Validation confirmed that IL1B, PTGS2, and SELL were robustly associated with both diseases. Immune landscape assessment demonstrated that these three genes exhibited significant negative correlations with regulatory T cells (Tregs) and positive correlations with neutrophil infiltration. Additionally, ten drugs with the highest predicted target specificity were identified. In conclusion, we utilized various machine learning and bioinformatics approaches to preliminarily elucidate potential comorbid mechanisms between AAA and periodontitis from a multidisciplinary perspective.

**Data availability statement:** The datasets used in this study (GSE10334, GSE23586, GSE57691, GSE16134 and GSE232911) are available from the GEO database.

**Funding:** This research was supported by research grants from the Medical Scientific Research Foundation of Zhejiang Province of China (2023RC106).

**Competing interests:** The authors have declared that no competing interests exist.

## 1. Introduction

The abdominal aorta is a major artery responsible for blood supply to the abdomen and lower limbs. Abdominal aortic aneurysm (AAA) is defined as an abnormal dilation exceeding 50% of the aorta's normal diameter, typically caused by vascular wall damage and intramural hematoma formation [1]. Common predisposing factors include hypertension, smoking, advanced age, and male sex [2]. Most AAA patients remain asymptomatic; however, aneurysm enlargement or rupture may lead to life-threatening complications such as internal hemorrhage and sudden death [3]. Studies have shown that the incidence of sudden death due to AAA rupture is 60–85%, and the incidence of AAA is continuously rising with aging populations, posing a serious threat to the health of elderly individuals worldwide [4]. Therefore, elucidating AAA pathogenesis, improving early diagnostic and therapeutic strategies, and discovering novel molecular biomarkers are critical priorities to prevent aneurysm progression and mitigate mortality risks.

Periodontitis is a chronic inflammatory disease of the periodontal tissues, primarily driven by dysbiotic microbial biofilms (dental plaque) and calculus deposition. Clinical manifestations include gingival swelling, bleeding, and progressive attachment loss, which may culminate in tooth loss in advanced stages [5]. Its pathogens can continuously invade the bloodstream during the chronic course of the disease, leading to cardiovascular diseases [6]. Established risk factors encompass smoking, diabetes, and age [7]. Emerging evidence indicates that periodontitis is associated with multiple systemic conditions, including intracranial aneurysms [8], Alzheimer's disease [9], and rheumatoid arthritis [10].

Although AAA and periodontitis are located in different anatomical sites, past studies have suggested a potential association between them [11]. Firstly, they share common risk factors, including smoking and age; however, the molecular mechanisms underlying this association remain unclear. Secondly, in a cross-sectional study, the authors studied periodontitis in 61 AAA patients, including 30 stable AAA patients and 31 unstable AAA patients, concluding that periodontitis had a high prevalence in both types of AAA patients. Porphyrin monobacterium was observed in samples of AAA patients with periodontitis, and it was correlated with AAA diameter and volume [12]. In another study, the authors suggested that AAA patients exhibited more severe periodontitis compared to patients unaffected by AAA [13].

Many opinions now believe that periodontitis is associated with systemic inflammatory response [14], and the presence and progression of abdominal aortic aneurysms are linked to inflammatory responses [15], which may be a mechanism for the association between the two. Since most previous studies have focused only on the prevalence and progression of periodontitis in AAA, research on potential interacting genes between these two diseases is remains unclear. Therefore, this study aims to explore and predict potential interacting genes between AAA patients and periodontitis through comprehensive bioinformatics strategies, and evaluate immune changes to provide reference for future prevention and treatment.

## 2. Materials and methods

### 2.1. Data source

Firstly, our workflow is illustrated in Fig 1. In this study, datasets related to AAA and periodontitis were sourced from the GEO database with the following search strategy: (1) search keywords were "periodontitis" and "abdominal aortic

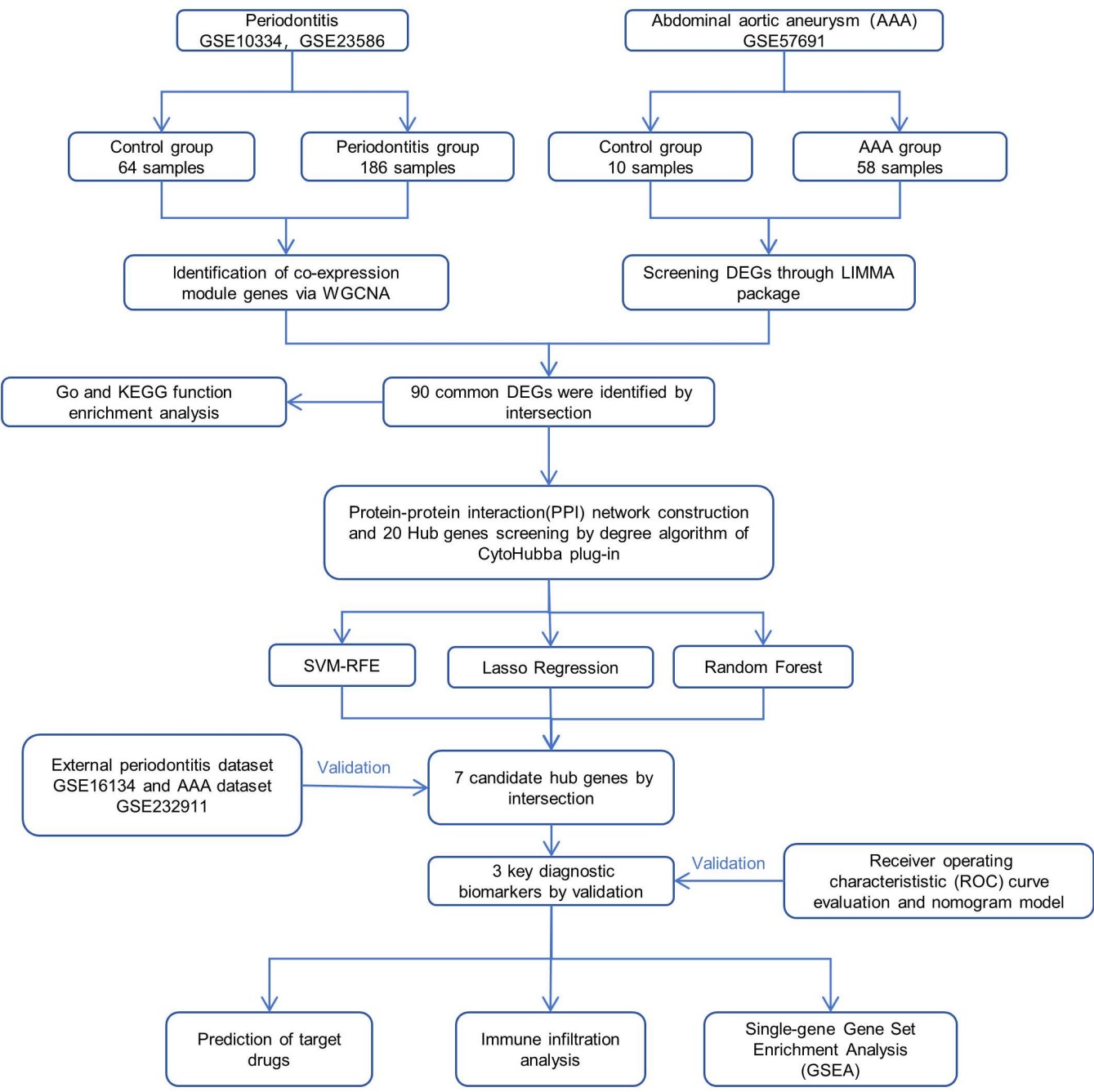

**Fig 1. Flowchart of the study design.**

aneurysm". (2) Samples used were from human-based array expression profiles. (3) The dataset sample size was ensured to be no less than 6 cases to guarantee the accuracy of the study results. As shown in Table 1, finally, we utilized five accessible datasets, including two datasets on AAA (GSE57691 [16] and GSE232911 [17]), with GSE57691 as the experimental set consisting of 10 control samples and 58 case samples, and GSE232911 as the validation set consisting of 26 control samples and 220 case samples. Regarding periodontitis, a total of 3 datasets were searched. We merged two available datasets (GSE10334 [18] and GSE23586 [19]) as the experimental set. To reduce batch effects in the combined sample sets of these two datasets, we performed batch correction using the R SVA package, resulting in 64 control samples and 186 case samples. Following this, We conducted principal component analysis (PCA) on the expression values of the samples both before and after batch correction for the merged dataset (S1 Fig). GSE16134 [20] was used as the validation set, comprising 69 control samples and 241 case samples.

## 2.2. Data preprocessing and differential expression analysis

Initially, for the extracted chip raw data, we utilized R software to convert probe IDs into gene symbols based on the corresponding probe platform information. In cases where multiple probes corresponded to a single gene, we calculated the average expression value of that gene across all samples. We used the R package "Limma" to analyze GSE57691 by setting the cutoff criteria as $|logFC| \geq 1.0$ and $p < 0.05$ to determine the differentially expressed genes (DEGs) for AAA. DEGs with $|logFC| \geq 2.0$ and $p < 0.05$ were also annotated.

## 2.3. Weighted gene co-expression network analysis

Initially, we calculated the Median Absolute Deviation (MAD) of the gene expression profiles from the corrected periodontitis dataset and excluded the bottom 50% of genes with the smallest MAD. In order to construct a co-expression network, we employed the "WGCNA" R package and utilized the GoodSamplesGenes method to remove outlier genes and samples [21]. More precisely, we computed the Pearson correlation matrix and applied an average linkage to all gene pairs. Subsequently, we constructed a weighted adjacency matrix using the power function $A\_mn = |C\_mn|^\beta$, where $\beta$ represents a soft-thresholding parameter that accentuates strong gene correlations while mitigating weaker ones. The soft-thresholding power ($\beta$) was set to 8 to satisfy scale-free topology (scale-free $R^2 > 0.85$) as recommended for signed networks. Module detection employed the dynamicTreeCut algorithm with a sensitivity parameter of 3 to ensure biologically meaningful cluster resolution, maintaining default height cutoffs of 0.99 and deepSplit = 2 [22]. These parameters optimize the balance between module granularity and gene assignment accuracy. Furthermore, we merged modules with a distance smaller than 0.25, resulting in a total of 18 co-expression modules. We then determined key modules by calculating the correlation between each module and the disease and gene significance for the selected modules of periodontitis (cyan, black, dark olive green) (S2 Fig). It's noteworthy that the grey module was considered a gene set that couldn't be assigned to any module.

**Table 1. Comprehensive details regarding the expression profile datasets utilized in this study.**

| GSE series No. | Type | Periodontitis | Healthy | Source tissue | Platform | Cohort |
|---|---|---|---|---|---|---|
| GSE10334 | mRNA | 183 | 64 | Gingival tissue | GPL570 | Experimental set |
| GSE23586 | mRNA | 3 | 3 | Gingival tissue | GPL570 | Experimental set |
| GSE16134 | mRNA | 241 | 69 | Gingival tissue | GPL570 | Validation set |
| GSE series No. | Type | AAA | Healthy | Source tissue | Platform | Cohort |
| GSE57691 | mRNA | 58 | 10 | Human abdominal aorta | GPL10558 | Experimental set |
| GSE232911 | mRNA | 220 | 26 | Human abdominal aorta | GPL17586 | Validation set |

AAA:abdominal aortic aneurysm.

## 2.4. Functional enrichment analysis of interacting genes

We performed enrichment analysis on the shared DEGs using the "clusterProfiler" R package, utilizing the most recent KEGG Pathway gene annotations retrieved from the KEGG rest API to acquire gene set enrichment results [23]. Subsequently, we performed enrichment analysis of overlapping DEGs using the gene ontology (GO) annotations from the "org. Hs.e.g.,db" R package with the "clusterProfiler" package to obtain enrichment results of gene sets [24]. A p-value of < 0.05 or a false discovery rate (FDR) of < 0.25 were considered statistically significant.

## 2.5. Construction of protein-protein interaction (PPI) network

We utilized the STRING online database to identify protein or gene interactions [25] by constructing a PPI network for interacting genes with a minimum required interaction score set to 0.4, and removed genes that couldn't interact, resulting in a PPI network containing 37 nodes and 51 edges. Following this, we conducted visualization using Cytoscape software, and identified the top 20 interacting genes utilizing the degree algorithm provided by the CytoHubba plugin.

## 2.6. Screening of candidate key interacting genes using 3 machine learning algorithms

To further refine the selection of candidate key interacting genes, we employed three machine learning algorithms, namely LASSO [26], SVM-RFE [27], and Random Forest (RF) [28], to calculate the top 20 interacting genes identified from the PPI network. Previous studies have widely employed machine learning methods for gene selection, including SVM-RFE, RF, and LASSO regression [29]. Candidate gene refinement was conducted using three machine learning algorithms with stratified 10-fold cross-validation repeated 5 times: (1) LASSO regression with penalty parameter selected via minimum Bayesian Information Criterion (BIC) under L1 penalty (alpha = 1); (2) SVM-RFE employing linear kernel, cost parameter C = 1, and recursive feature elimination until 9 features; and (3) Random Forest configured with ntree = 500 and mtry = sqrt(p) (where p represents feature count) [30]. These hyperparameter settings followed established bioinformatics protocols, and model performance was rigorously evaluated by mean area under the curve (AUC) computed across 50 validation cycles.

## 2.7. Validation of key interacting genes using independent external datasets

To enhance the accuracy of this study, we validated the relative expression levels and trends of candidate key interacting genes using the "GSE232911" AAA validation set and the "GSE16134" periodontitis validation set. Through this validation, we further confirmed the key interacting genes.

## 2.8. Construction and validation of receiver operating characteristic (ROC) curves and nomograms

Initially, we assessed the diagnostic value of the candidate genes by plotting ROC curves and calculating the area under the curve (AUC) along with the corresponding 95% confidence interval (CI) [31]. Typically, an AUC > 0.6 is considered diagnostically valuable for the selected genes. Subsequently, we constructed nomograms using the "rms" R package. Lastly, we quantified the ROC of the nomograms to further predict the diagnostic value of the diagnostic model [32].

### 2.8.1. Immune infiltration analysis.
Initially, we used the "cibersort" algorithm to analyze the relative expression levels of different immune cells between the two sets of control and case test datasets [33]. Finally, we calculated the correlation between the expression of validated key interacting genes and different immune cell expressions. A p-value < 0.05 was considered statistically significant, and visualization was performed using the "ggplot2" R package.

### 2.8.2. Single-gene gene set enrichment analysis (GSEA) and drug prediction.
Initially, we employed the "GSEA" R package to compute the correlation between each key interacting gene and other genes [34]. Subsequently, to further explore their potential pathological mechanisms, we assessed the enrichment levels of signaling pathways through GSEA for the three genes. Additionally, we predicted potential drugs for key interacting genes through the "Enrichr" database [35], for the treatment or prevention of AAA.

## 3. Results

### 3.1. Identification of DEGs in AAA

As shown in Fig 2A, B, in the GSE57691 dataset, we utilized the "Limma" R package to identify a total of 323 DEGs between AAA and the control group, including 55 upregulated and 268 downregulated genes (criteria: |logFC| ≥ 1.0, p < 0.05 compared to healthy control group), and DEGs with |logFC| ≥ 2.0 were annotated.

### 3.2. Construction of weighted gene co-expression networks for periodontitis and selection of key modules

In briefly, based on a soft-thresholding parameter of 8 (Fig 3A, B), WGCNA analysis identified a total of 18 co-expression modules (Fig 3C) in the combined periodontitis dataset (GSE10334 + GSE23586). Finally, based on the module-phenotype heatmap (Fig 3D), the darkolivegreen module (63 genes, correlation coefficient (CC) = −0.44, p < 0.0001) and black module (2920 genes, CC = −0.43, p < 0.0001) showed the largest negative correlations, while the cyan module (1429

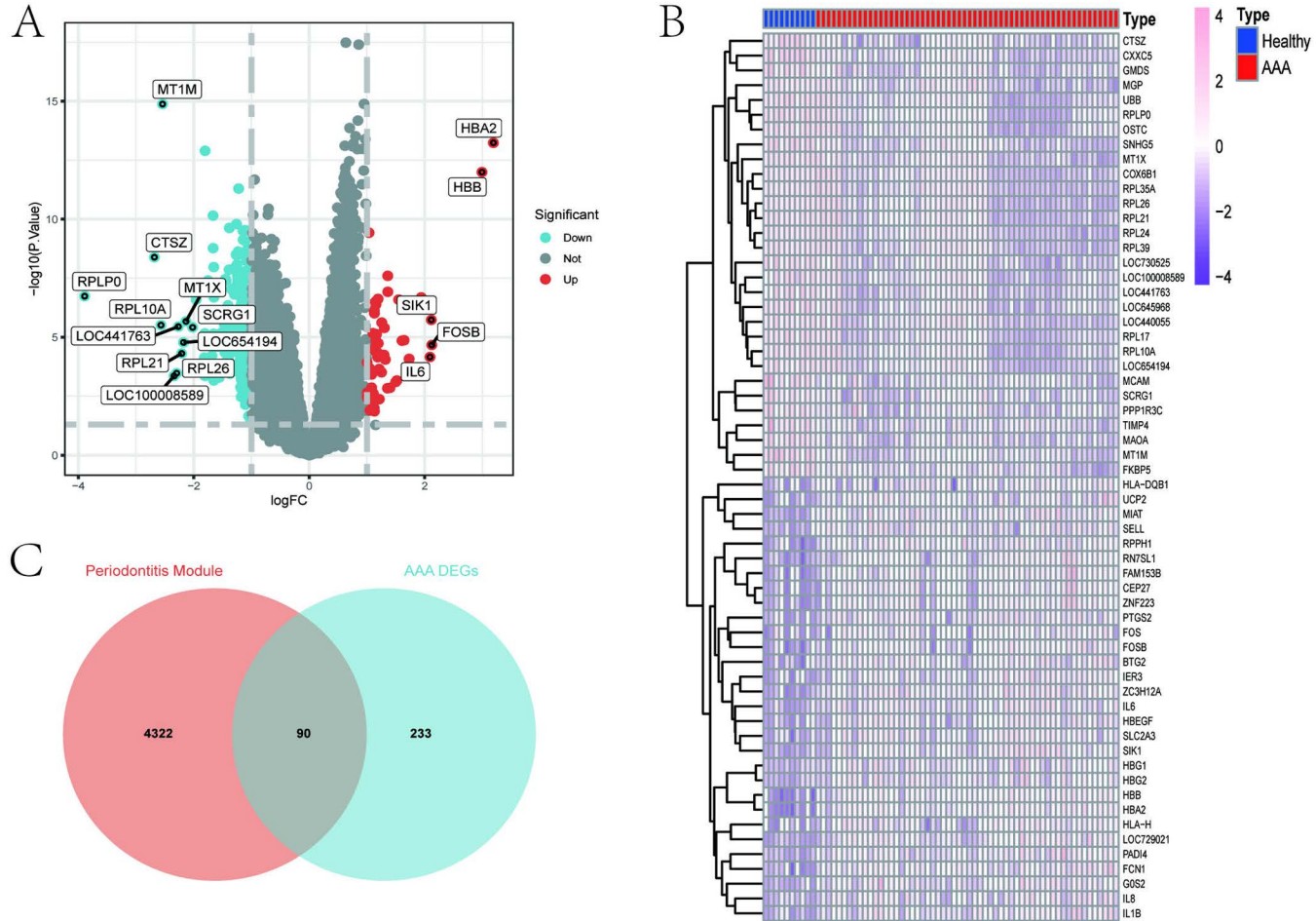

**Fig 2. From the Abdominal Aortic Aneurysm (AAA) dataset (GSE57691), DEGs between the AAA group and the control group were identified.** (A) Volcano plot of all genes, where red and blue dots respectively represent 55 upregulated and 268 downregulated genes (|logFC| ≥ 1.0, p < 0.05), with genes having |logFC| ≥ 2.0 marked. (B) Heatmap of 60 upregulated and downregulated DEGs, with pink and blue grids representing upregulated and downregulated DEGs, respectively. (C) Venn diagram displaying the intersection of Weighted Gene Co-expression Network Analysis (WGCNA) important module genes associated with periodontitis and the identified DEGs of AAA mentioned above.

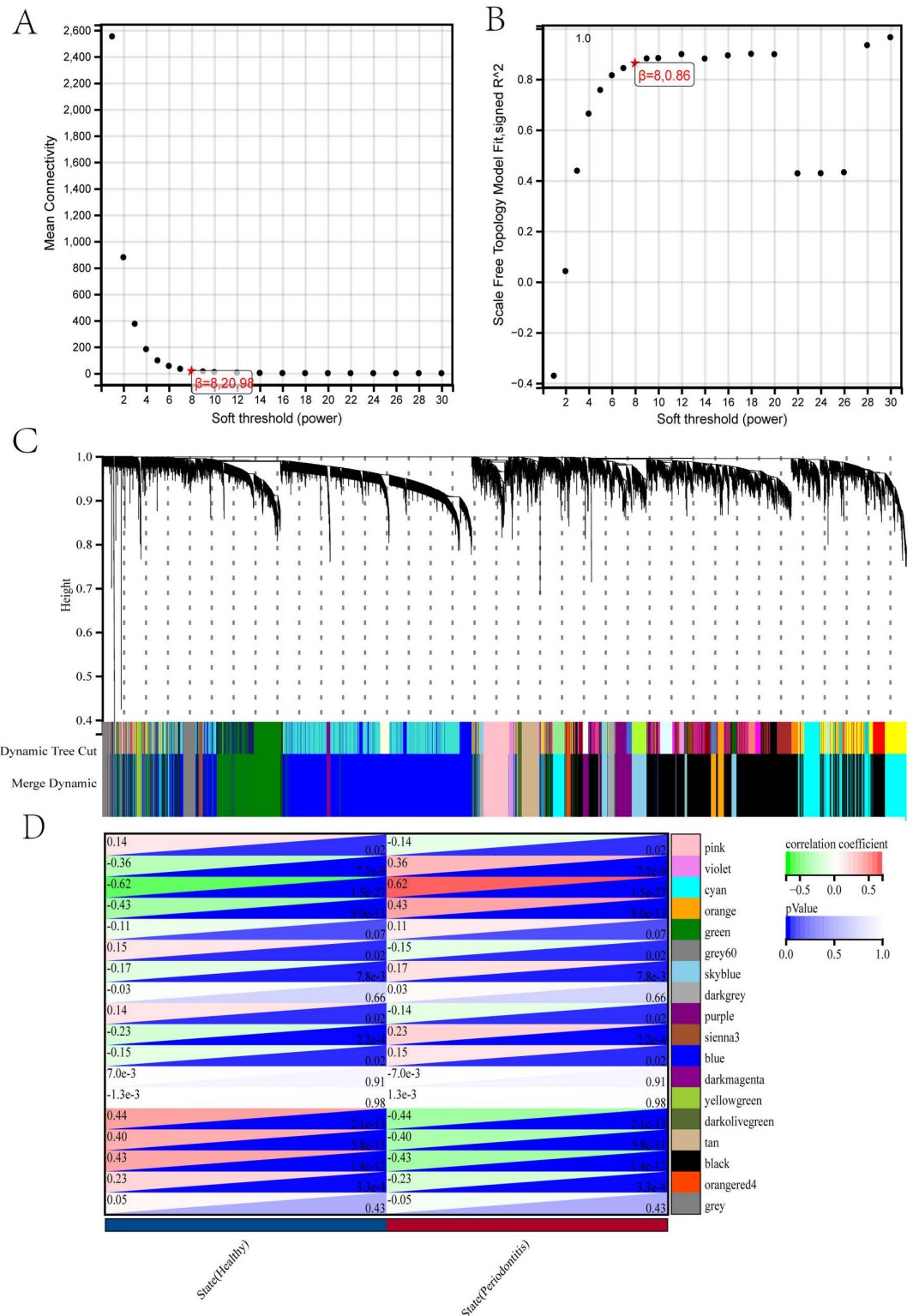

**Fig 3. WGCNA was employed to identify modules associated with the synthetic periodontitis dataset.** (A, B) Based on scale independence and average connectivity, β = 8 was considered the optimal soft-thresholding value. (C) Gene clustering tree with multiple partitioned modules. Different clusters are connected by different colors. (D) Heatmap of adjacency of feature genes.

genes, CC = 0.62, p < 0.0001) exhibited the greatest positive correlation. Therefore, the genes screened out were included in the subsequent study and make scree plot and significance scatter plot of module membership.

### 3.3. Functional enrichment analysis of interacting genes

Initially, we plotted a Venn diagram of interacting genes related to AAA and periodontitis using the "ggvenn" package (Fig 2C), which revealed 90 overlapping interacting genes. Subsequently, we performed functional enrichment analysis on these 90 interacting genes, including GO and KEGG analyses, to understand the potential underlying mechanisms. As shown in Fig 4A–D, GO analysis revealed that these interacting genes were mainly enriched in several aspects: (1) biological processes (BP), including cell migration, cell adhesion, and response to nitrogen compounds(Fig 4A); (2) cellular components (CC), including extracellular matrix, sarcomere, growth factor complex, and insulin-like growth factor binding protein complex(Fig 4B); (3) molecular functions (MF), including CCR5 chemokine receptor binding, among others(Fig 4C). KEGG enrichment analysis results indicated that these interacting genes were mainly enriched in pathways such as arachidonic acid metabolism, tryptophan metabolism, regulation of actin cytoskeleton, and fluid shear stress and atherosclerosis(Fig 4D). The above results suggest that the potential comorbid mechanisms between AAA and periodontitis involve immune, metabolic, and inflammatory aspects.

### 3.4. Further screening of interacting genes using PPI network

To further screen key interacting genes between AAA and periodontitis, we constructed a PPI network with 37 nodes and 51 edges after excluding 53 non-interacting genes, as visualized in Fig 4E. Subsequently, we utilized the Degree algorithm in the CytoHubba plugin to rank and further select the top 20 interacting genes, as shown in Fig 4F.

### 3.5. Further screening of candidate key interacting genes using three machine learning algorithms

Illustrated in Fig 5A–G, we initially utilized LASSO, SVM-RFE, and RF machine learning algorithms to further refine the selection of the 20 interacting genes. In LASSO regression, we observed that 11 interacting genes reached the minimum Bayesian Information Criterion (BIC) on the curve (Fig 5A, B). For SVM-RFE, the initial 9 genes demonstrated the highest accuracy and the lowest prediction error concerning the relationship between AAA and periodontitis (Fig 5C, D). Meanwhile, with RF, we identified the top 10 genes consistent with their importance scores (Fig 5E, F). Finally, we intersected the outcomes from the three machine learning methods and constructed a Venn diagram, thereby identifying 7 candidate key interacting genes (IL1B, PTGS2, SELL, ROCK2, IGFBP3, MYH10, DDIT4) (Fig 5G).

### 3.6. Validation and determination of candidate key interacting genes, and establishment of a nomogram

Firstly, we validated the relative expression levels and trends of the aforementioned 7 candidate key interacting genes using an external independent dataset. As shown in Fig 6A–D, 3 candidate interacting genes (IL1B, PTGS2, SELL) exhibited consistent trends and meaningful relative expression levels, suggesting them as key interacting genes. Following this, to evaluate the diagnostic significance of these three key interacting genes, ROC curves were generated based on the individual characteristics and expressions of each gene in both the testing and validation datasets (Fig 7A–L). As shown in Table 2, we can clearly observe the AUC and 95% CI of these 3 genes in each group. Based on the ROC validation, we considered these 3 genes to have diagnostic value. Subsequently, by quantifying the relative expression of each gene, we constructed a nomogram to evaluate the effectiveness of these 3 key interacting genes in disease diagnosis, and further validated them using ROC curves (AUC 0.89, 95% CI 0.84–0.94) (Fig 8A, B).

### 3.7. Single-gene GSEA of key interacting genes

To further explore the potential KEGG pathways in which these genes are involved in the shared pathogenic mechanisms of AAA and periodontitis, we performed GSEA on IL1B, PTGS2, and SELL. As shown in Fig 8C–E, each gene exhibited

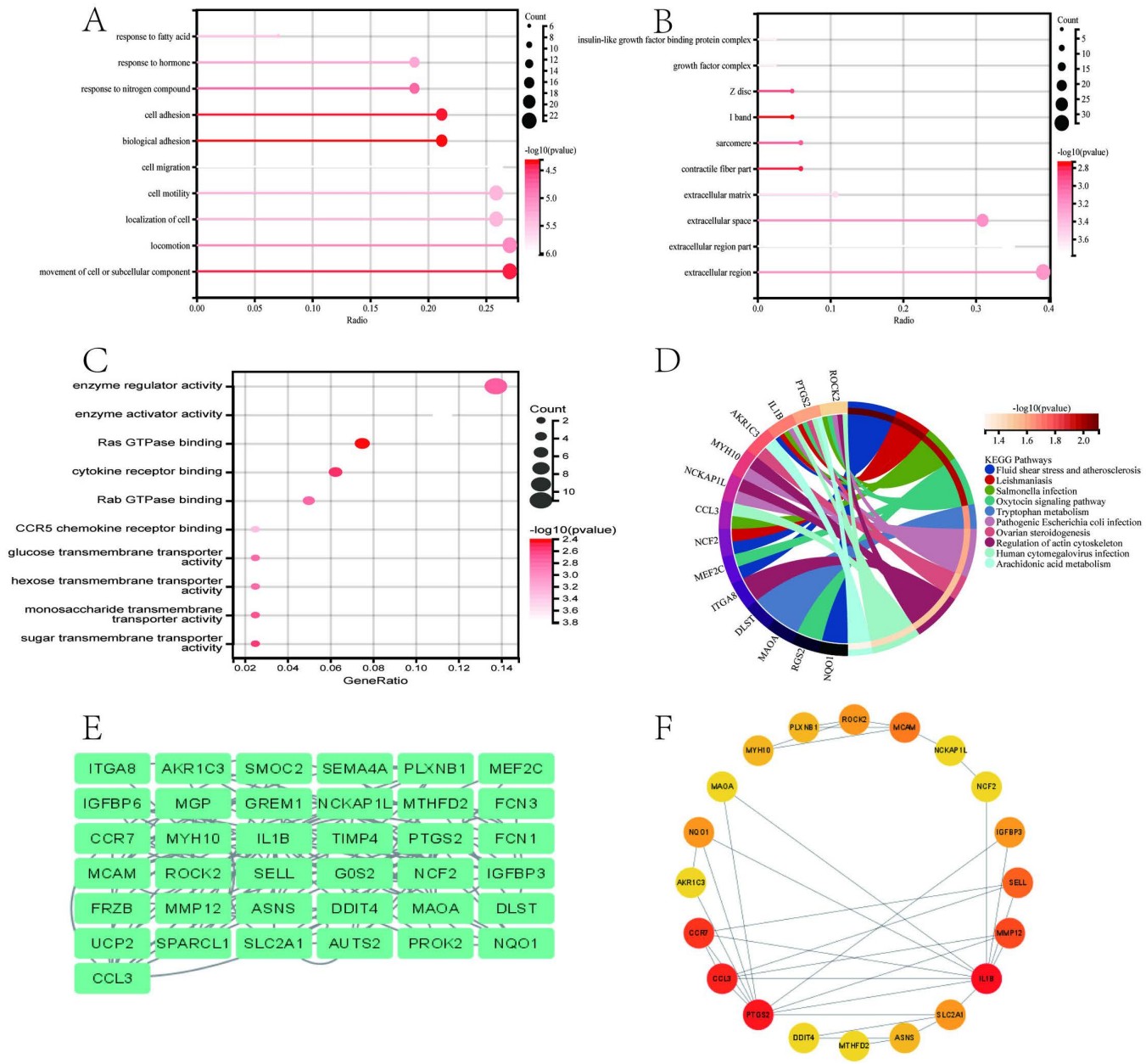

**Fig 4. Functional enrichment analysis of periodontitis-related DEGs in AAA and identification of candidate key hub genes through Protein-Protein Interaction (PPI) network were conducted.** (A-C) Gene Ontology (GO) functional analysis of periodontitis-related DEGs in AAA, including Biological Processes (BP), Cellular Components (CC), and Molecular Functions (MF). The x-axis and y-axis represent gene ratio and GO terms, respectively; the color of circles indicates significance, and the size represents the number of enriched genes. (D) Kyoto Encyclopedia of Genes and Genomes (KEGG) pathway analysis of periodontitis-related DEGs in AAA. (E, F) A PPI network of 37 interacting genes was selected using the "degree" algorithm, and the top 20 interacting genes were visualized in Cytoscape as a network.

10 significant pathways. For instance, IL1B was involved in mismatch repair, PTGS2 participated in the IL-17 signaling pathway and TNF signaling pathway, SELL was associated with autoimmune thyroid disease, and notably, all three genes were related to tyrosine metabolism. Based on these results, we infer that these three genes may be involved in

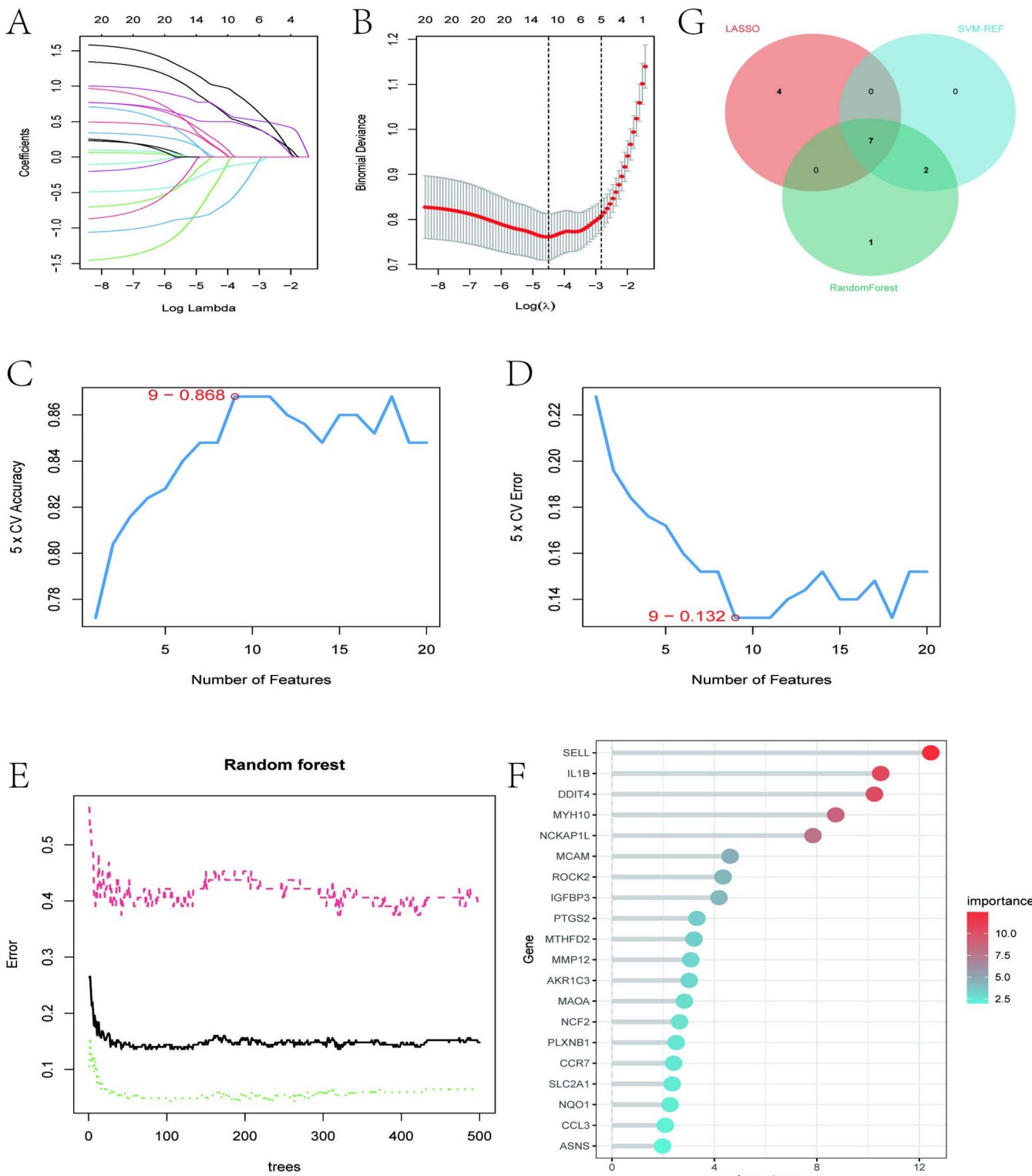

**Fig 5. Three machine learning algorithms were further employed to select candidate key hub genes.** (A, B) Based on 10-fold cross-validation, the minimum absolute shrinkage and selection operator (LASSO) regression model feature selection identified 11 genes corresponding to the lowest point of the curve, considered optimal biomarkers for studying the shared pathogenic mechanisms of AAA and periodontitis. (C, D) The Support Vector Machine Recursive Feature Elimination (SVM-RFE) algorithm was used to select significant feature genes from the 20 candidate genes. Nine genes (maximum

accuracy = 0.868, minimum error rate = 0.132) were ultimately identified as optimal feature genes. (E, F) Gene selection based on the Random Forest (RF) calculation strategy listed the top 10 genes according to their importance scores. (G) RF, SVM-RFE, and LASSO cross-regression were combined, and a Venn diagram was visualized, resulting in 7 candidate key hub genes (IL1B, PTGS2, SELL, ROCK2, IGFBP3, MYH10, DDIT4) for further validation.

the shared pathogenic mechanisms of AAA and periodontitis through aspects such as pathogen invasion, metabolism, immune response, and inflammation.

### 3.8. Immunological landscape analysis

As shown in Fig 9A–D and S3 C, D Fig, we first visualized the relative percentages of 16 immune cells in the case and control groups of AAA and periodontitis using stacked bar plots, box plots, and heatmaps. The results revealed significant changes in seven meaningful immune cell types in the case group of AAA (T cells follicular helper, regulatory T cells (Tregs), Monocytes, Macrophages M1, Macrophages M2, Dendritic cells activated, Neutrophils), while the immune landscape in the case group of periodontitis showed significant alterations compared to the control group. Furthermore, the gene correlation heatmap indicated that the expression levels of IL1B, PTGS2, and SELL were significantly negatively correlated with T cells CD8, regulatory T cells (Tregs), Mast cells resting, Dendritic cells resting, and significantly positively correlated with Neutrophils(Fig 8F). Therefore, we have reason to believe that regulatory T cells (Tregs) and Neutrophils may play a role in the potential shared pathogenic mechanisms between AAA and periodontitis.

### 3.9. Targeted drug prediction

As depicted in S3A, B Figs, based on the prediction results from the Enrichr online database, we found that ten drugs (flurbiprofen, nimesulide, Calcimycin, thalidomide, 2'-Hydroxychalcone, Rev 5901, 1-BROMOPROPANE, meloxicam, methanol, ILOPROST) were predicted to be the most likely targeted drugs for periodontitis and AAA (S3B Fig). Among these, PTGS2 was the most relevant biomarker associated with these drugs (S3A Fig), suggesting that PTGS2 has significant potential in the prevention and treatment of AAA and could be a target for drug research. As detailed in Table 3.

## 4. Discussion

Current research indicates that periodontal disease is associated not only with the occurrence and rupture of intracranial aneurysms [36], atherosclerotic cardiovascular disease [37], and venous thromboembolism [38], but also with AAA [39]. Mechanistically, periodontal pathogens such as Porphyromonas gingivalis induce chronic gingival inflammation and tissue destruction through dysregulated host-microbe interactions [40]. The sustained low-grade inflammation triggered by periodontal disease may induce systemic inflammatory responses, rendering the vascular wall more susceptible to inflammation and damage, thereby increasing the risk of AAA [41]. Moreover, certain common inflammatory mediators and biomarkers may play connecting roles between periodontal disease and AAA. However, the specific mechanisms by which periodontal disease affects abdominal aortic aneurysms have not been fully studied. In this study, we focused on exploring the key crosstalk genes shared by AAA and periodontal disease (IL1B, PTGS2, and SELL) through comprehensive bioinformatics methods and validated using an external independent dataset.

The IL1B gene encodes interleukin-1β (IL-1β), a key pro-inflammatory cytokine essential for regulating biological functions including inflammatory responses, immune reactions, and cell proliferation [42]. It is of significant importance in the immune system's responses, particularly in the initiation and advancement of inflammatory conditions [43]. Additionally, abnormal expression or excessive activation of IL-1β is associated with the onset and progression of various diseases such as rheumatoid arthritis [44], inflammatory bowel disease [45], among others. In a study comprising 34 AAA subjects and 34 non-AAA controls, significantly elevated plasma levels of IL-1α and IL-1β were observed in AAA subjects

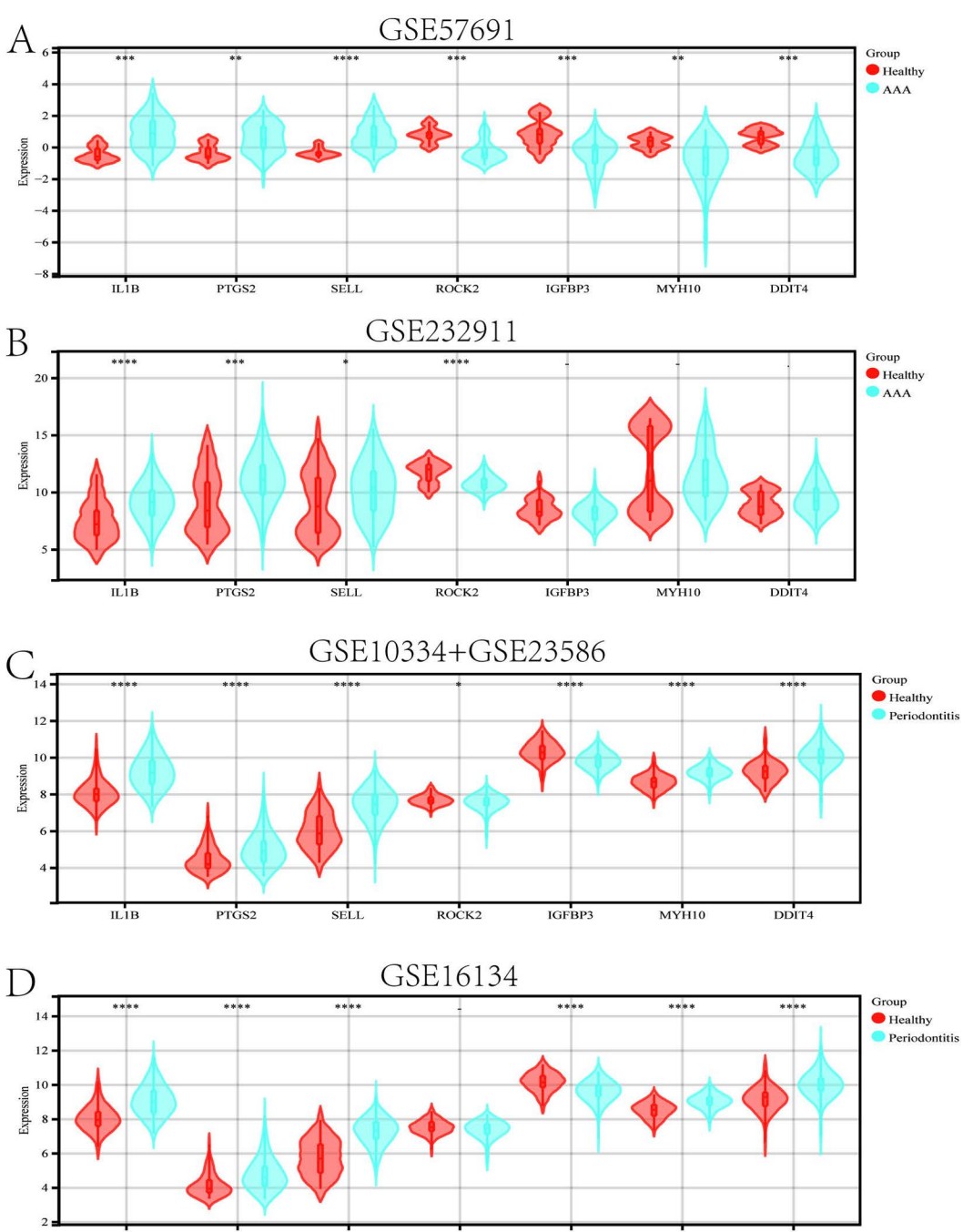

**Fig 6. The relative expression levels of the 7 candidate key hub genes (IL1B, PTGS2, SELL, ROCK2, IGFBP3, MYH10, DDIT4) in the test and validation datasets of AAA and periodontitis were compared.** (A, B) Comparison of expression levels of candidate genes in the AAA test dataset (GSE57691) and validation dataset (GSE232911). (C, D) Comparison of expression levels of candidate genes in the periodontitis validation dataset (GSE10334+GSE23586) and validation dataset (GSE16134).- p > 0.05; * p < 0.05; ** p < 0.01; *** p < 0.001; **** p < 0.0001.

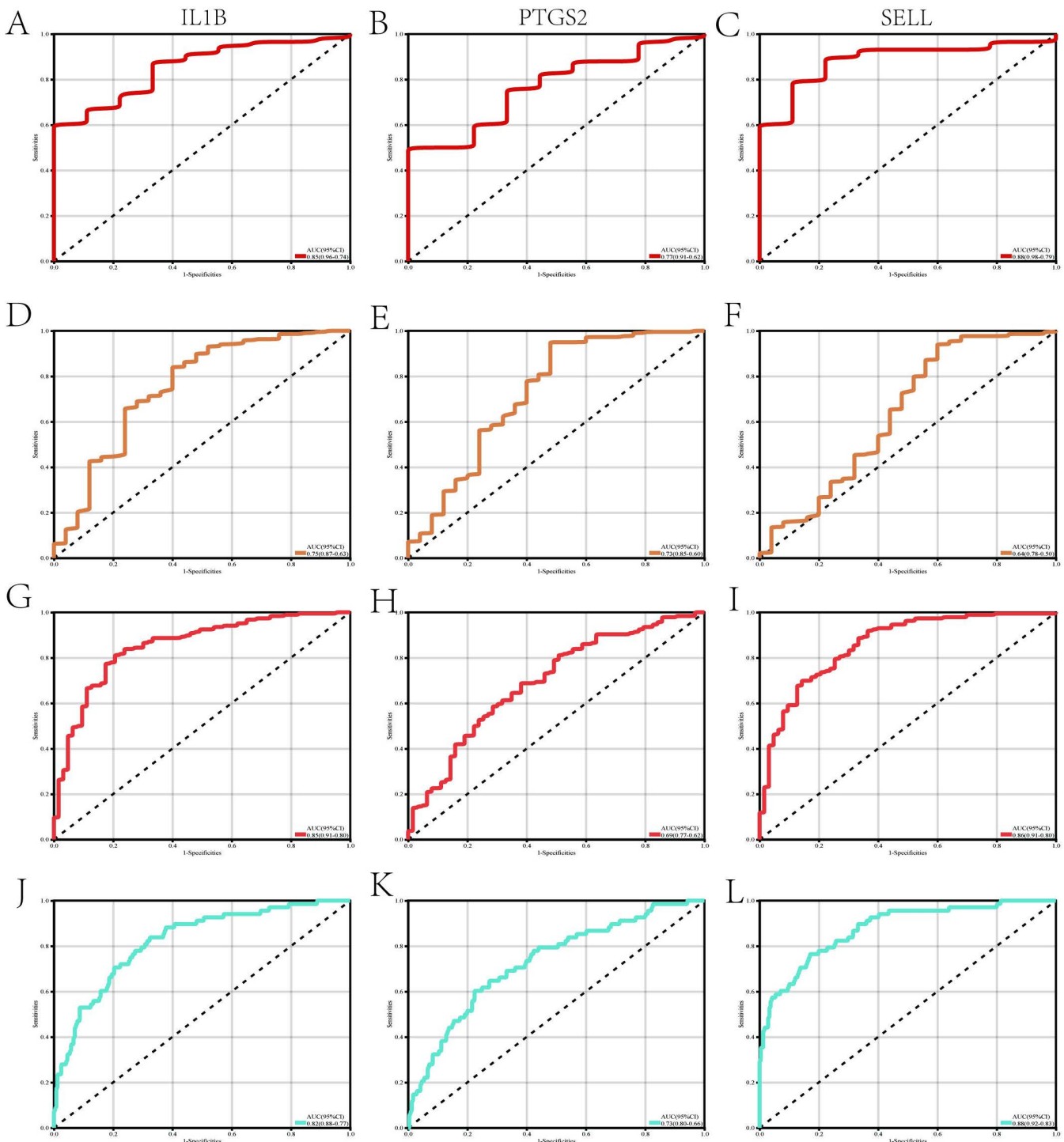

**Fig 7. Evaluation of the predictive capability of key hub genes.** (A-C) ROC curves of three key hub genes (IL1B, PTGS2, SELL) in the AAA test dataset (GSE57691), (D-F)ROC curves of three key hub genes (IL1B, PTGS2, SELL) in the AAA validation dataset (GSE232911).(G-I) ROC curves of three key hub genes in the periodontitis test dataset (GSE10334+GSE23586), (J-L)ROC curves of three key hub genes in the periodontitis validation dataset (GSE16134).

**Table 2. Detailed information of ROC curves in the study.**

| | Periodontitis | | | AAA | | |
|---|---|---|---|---|---|---|
| | Gene name | AUC | 95%CI | Gene name | AUC | 95%CI |
| Experimental set | IL1B | 0.85 | 0.80-0.91 | IL1B | 0.85 | 0.74-0.96 |
| | PTGS2 | 0.69 | 0.62-0.77 | PTGS2 | 0.77 | 0.62-0.91 |
| | SELL | 0.86 | 0.80-0.91 | SELL | 0.88 | 0.79-0.98 |
| Validation set | IL1B | 0.82 | 0.77-0.88 | IL1B | 0.75 | 0.63-0.87 |
| | PTGS2 | 0.73 | 0.66-0.80 | PTGS2 | 0.73 | 0.60-0.85 |
| | SELL | 0.88 | 0.83-0.92 | SELL | 0.64 | 0.50-0.78 |

ROC:Receiver Operating Characteristic, AAA:abdominal aortic aneurysm.

compared to the control group. Furthermore, the expression levels of IL1B mRNA in peripheral blood mononuclear cells from male AAA patients were significantly higher than in female patients (2.34 vs. 0.25, P < 0.000001) [46]. It is noteworthy that in an Angiotensin II (AngII)-induced AAA mouse model, Isoda et al. demonstrated significant inhibition of AAA formation in mice by suppressing IL-1β [47]. Additionally, Johnston et al. demonstrated that AAA formation induced by elastase could be weakened by gene deletion or receptor antagonism of IL-1β [48]. Regarding periodontal disease, a study involving 186 periodontitis patients and 208 control subjects demonstrated an association between IL1B gene polymorphisms and susceptibility to PD [49]. The significant upregulation of Interleukin-1 beta (IL1B) in both diseases highlights its role as a master inflammatory regulator. In periodontitis, IL1B is a key effector of the host response to periodontal pathogens (e.g., Porphyromonas gingivalis) [50]. Its release is triggered through NLRP3 inflammasome activation in macrophages, driving gingival tissue destruction and alveolar bone resorption via induction of matrix metalloproteinases (MMPs) and receptor activator of nuclear factor kappa-B ligand (RANKL) [51]. In AAA, IL1B promotes vascular smooth muscle cell (VSMC) apoptosis and extracellular matrix degradation by upregulating MMP-2/9 expression in infiltrating macrophages and neutrophils [52]. Thus, IL1B serves as a pathogenic nexus that propagates inflammatory damage across distant sites.

The gene PTGS2, also known as cyclooxygenase-2 (COX-2), encodes an enzyme involved in prostaglandin synthesis, particularly the synthesis of prostaglandin E2 (PGE2), an important pro-inflammatory mediator [53]. PTGS2 plays a crucial role in processes such as inflammation, cell proliferation, and angiogenesis. Its overexpression is associated with the onset and development of various diseases, including inflammatory conditions [54] and cancer [55]. Gan et al. identified that the activation of miR-15b-5p results in decreased expression levels of ACSS2 and its downstream gene PTGS2, consequently fostering apoptosis of vascular smooth muscle cells (VSMCs) and impeding their proliferation. They tentatively suggested the miR-15B-5P/ACSS2/PTGS2 axis as a promising therapeutic target for addressing AAA [56]. Zhang et al. identified six key genes (CCL2, CCR7, CXCL1, CXCL8, PTGS2, and SELL) involved in the pathogenesis of AAA through differential analysis of GEO datasets [57]. In a survey involving 60 patients with chronic periodontitis and 60 control subjects, the levels of PTGS2 in plasma samples were measured. The results indicated a statistically significant elevation in plasma PTGS2 concentration in patients with periodontitis compared to the control group (p = 0.001) [58]. Caetano et al. evaluated disease-associated genes identified by GWAS related to periodontitis and analyzed the expression of IL1B, PTGS2, FCGR2A, IL10, and IL1A in the bone marrow of four periodontitis patients. They found that IL1B and PTGS2 exhibited the highest expression specificity compared to other genes, and IL1B and PTGS2 were involved in the inflammatory response [59]. The above research corroborate the miR-15b-5p/ACSS2/PTGS2 axis in AAA pathogenesis [56], while periodontitis studies confirm COX-2 inhibition reduces alveolar bone loss [60]. The conserved COX-2/PGE$_2$ axis thereby bridges tissue-specific destruction through shared pro-catabolic signaling.

The gene SELL (Selectin L) encodes a cell surface adhesion molecule belonging to the selectin family. Selectin L is primarily expressed on the surface of white blood cells and is involved in the adhesion and migration processes of these

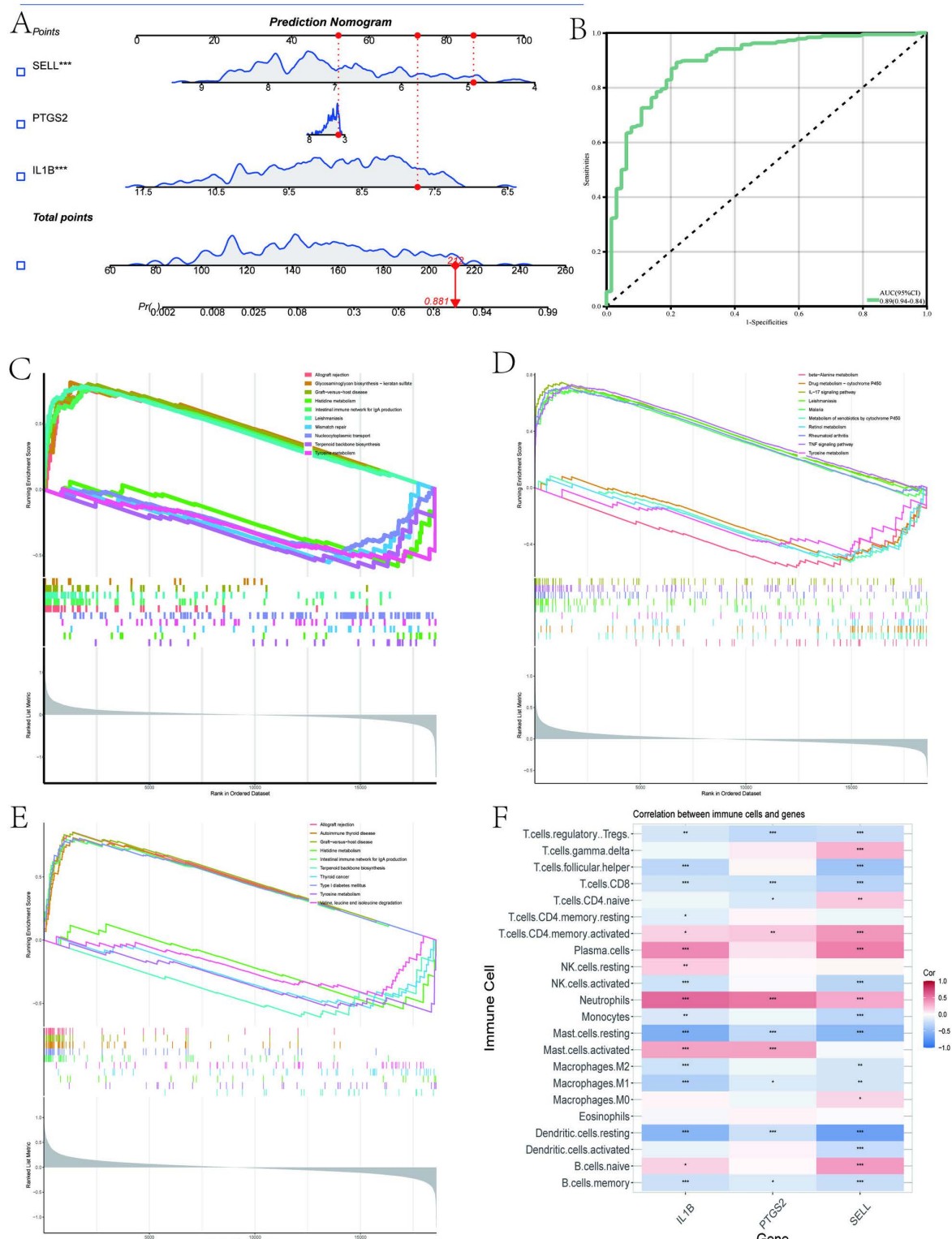

**Fig 8. Based on the three key hub genes, a nomogram model was constructed, and Single-gene Gene Set Enrichment Analysis (GSEA) was performed.** (A) Construction of a nomogram model based on three genes (IL1B, PTGS2, SELL). (B) Nomogram ROC diagnostic curve. (C) Single-gene GSEA of IL1B. (D) Single-gene GSEA of PTGS2. (E) Single-gene GSEA of SELL. (F) Heatmap showing the correlation between the three key genes and immune cells.

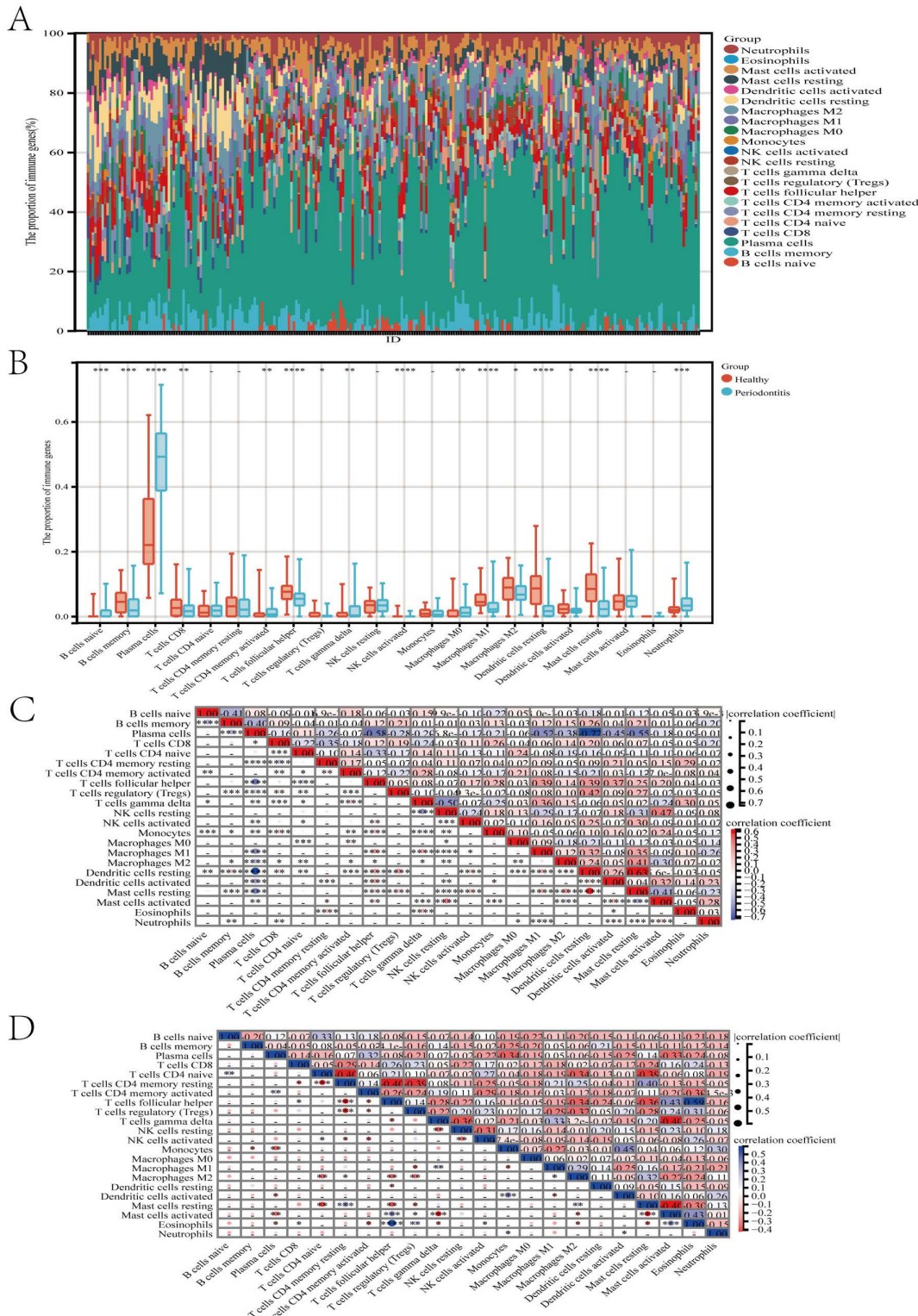

**Fig 9. Immunoscape analysis based on the three key hub genes.** (A, B, C) Box plots, stacked plots, and heatmaps displaying the relative proportions of 16 immune cells in different samples of the periodontitis dataset. (D) Heatmaps displaying the relative proportions of 16 immune cells in different samples of the AAA dataset. - p > 0.05; * p < 0.05; ** p < 0.01; *** p < 0.001; **** p < 0.0001.

**Table 3. Gene-drug interactions of key hub genes.**

| Gene | Targeted Drug |
|---|---|
| PTGS2 | flurbiprofen, nimesulide, Calcimycin, thalidomide, 2'-Hydroxychalcone, Rev 5901, 1-BROMOPROPANE, meloxicam |
| IL1B | flurbiprofen, nimesulide, Calcimycin, thalidomide, 2'-Hydroxychalcone, Rev 5901, 1-BROMOPROPANE |
| SELL | flurbiprofen, nimesulide, Calcimycin, thalidomide, meloxicam |

cells. It plays a crucial role in inflammation, immune response, and the circulation of white blood cells [61]. The expression level of Selectin L can be regulated by inflammation and immune activation, affecting the adhesion and migration of white blood cells, thereby regulating the inflammatory response and immune response. Pu et al. used microarray analysis to investigate AAA and demonstrated that a critical miRNA, miR-145, exerts an effect on AAA by targeting RAC2 and SELL [62]. In a recent study, they identified ITGAL and SELL as novel target genes for AAA and speculated that they might affect AAA through immune cells such as NK cells and Tregs [63]. Liu et al. identified oxidative stress-related genes for periodontitis through dataset analysis (CXCR4, SELL, FCGR3B, FCGR2B, PECAM1, and ITGAL) [64]. Furthermore, Pan et al. analyzed datasets and designed a study involving 120 patients with periodontitis and 40 control subjects. The results indicated a correlation between the expression level of the SELL gene and the severity of periodontitis [65]. In our study, IL1B, PTGS2, and SELL were significantly upregulated in both AAA and periodontitis compared to the control group. They may be involved in the comorbid mechanisms of AAA and periodontitis, although their specific mechanisms remain unclear, warranting further investigation.

As is well known, chronic inflammation drives the development of AAA, leading to continuous remodeling and separation of the arterial wall [66]. Periodontitis, being a chronic inflammatory disease, partially provides a premise for the occurrence of AAA, thereby increasing the risk of patients developing AAA. In our study, GO analysis and KEGG analysis indicated that these interacting genes mainly enrich aspects of immunity and inflammation. Arachidonic acid is a fatty acid and a key regulatory factor in inflammatory responses. When tissues are damaged or infected, arachidonic acid is further metabolized into prostaglandins and leukotrienes, among other steroid hormones, thereby participating in the regulation and transmission of inflammatory responses, leading to the migration of inflammatory cells and the release of inflammatory mediators [67]. Tryptophan is the starting material of the tryptophan pathway. This pathway involves the participation of immune-active factors, the metabolites of tryptophan, in immune regulation and the inflammatory process [68]. This includes the IL-17 signaling pathway [69] and the TNF signaling pathway [70] mentioned in subsequent single-gene GSEA, which are closely related to inflammation. Subsequently, we applied the CIBERSORT algorithm and identified the specificity of Tregs and neutrophils. There may be potential shared pathogenic mechanisms in their shared pathogenic mechanisms. Tregs are an important subset of immune cells, whose main function is to regulate and suppress the activity of the immune system to maintain immune balance and self-immune tolerance [71]. Studies have shown that Tregs can be detected in abdominal aortic aneurysms and atherosclerotic tissues, and their role is to prevent the development of atherosclerosis [72]. Liu et al. demonstrated that Tregs can inhibit the further development of AAA by suppressing the expression of COX-2, both in vitro and in vivo [73]. Some studies on periodontitis have confirmed the crucial role of Treg inhibitory in the late stages of periodontitis. For example, in one study, they induced periodontitis with Actinomyces and then inhibited Tregs with anti-GITR, resulting in increased alveolar bone loss [74]. The significant negative correlation between key hub genes (IL1B, PTGS2, SELL) and Tregs infiltration indicates impaired immunoregulation. Tregs normally suppress effector T cells and antigen-presenting cells through the secretion of IL-10 and TGF-β [75] and CTLA-4-mediated [76] dendritic cell inhibition. Research indicates that IL-1B promotes STAT3 phosphorylation, thereby inducing the transformation of Treg cells into Th17 cells [77]. Consequently, the resulting Treg suppression deficit permits uncontrolled activation of osteoclast precursors (leading to alveolar bone resorption in periodontitis) [78] and MMP-producing

macrophages (causing medial layer degradation in AAA) [79]. This mechanistic link explains the pathological significance of the observed correlation. Neutrophils are an important component of the innate immune system and play a crucial role in infection and inflammation processes. One study found that the occurrence of experimental AAA was significantly inhibited by depleting neutrophils in mice [80]. Additionally, a review mentioned the crucial role of neutrophils in the pathogenesis of periodontitis, and proposed some new insights, such as when there is a defect in white blood cell adhesion in periodontitis, the progression of periodontitis may be more related to host response dysregulation than to aggravated infection [81]. The aforementioned results suggest the potential involvement of both innate and regulatory immune responses in the common mechanisms underlying of AAA and periodontitis. Thus, by elucidating the interplay between the identified biomarkers (IL1B, PTGS2, and SELL) and immune cells (Tregs and neutrophils), a more comprehensive comprehension of the potential comorbid mechanisms underlying AAA and periodontitis can be attained. Drug prediction confirmed PTGS2's druggable nature; however, therapeutic targeting of PTGS2 presents a clinical paradox. while topical flurbiprofen significantly reduces gingival crevicular fluid PGE2 (>60%) and attenuates alveolar bone loss in periodontitis [12], systemic NSAID use is contraindicated in abdominal aortic aneurysm (AAA) due to suppression of vasoprotective prostacyclin (PGI2) and promotion of intra-luminal thrombosis [32]. This mechanistic dichotomy necessitates risk-stratified approaches: periodontitis-only patients may benefit from localized NSAID delivery (0.3% flurbiprofen gel) without systemic exposure [12]; AAA-affected individuals require strict avoidance of NSAIDs given accelerated aneurysm expansion rates (+0.8 mm/year) in users [82]; and high-risk comorbid cases should prioritize IL1B blockade via canakinumab (150 mg subcutaneously every 3 months), which demonstrated 15% cardiovascular event reduction in the CANTOS trial without compromising vascular integrity [83]. Thus, although PTGS2 constitutes a mechanistically compelling target, its pharmacologic inhibition carries AAA-specific contraindications that redirect therapeutic focus toward upstream modulators like IL1B.

This study identified key interacting genes in AAA and periodontitis from the perspective of immunity and inflammation, providing new insights into the comorbid mechanisms of AAA and periodontitis. However, this study also has certain limitations. Firstly, we did not stratify the disease data collected from the GEO dataset, which may lead to some bias in the results. Secondly, although our bioinformatics analysis provides valuable insights, there is no experimental data to validate the function of these genes and their role in the crosstalk between AAA and periodontitis. Therefore, future research should consider experimental validation of the expression levels and interactions of these genes in clinical samples to confirm their specific roles in the disease processes. Additionally, although we predicted targeted drugs for key interacting genes and selected 10 drugs most likely to target these genes, our drug prediction focused on target affinity without incorporating clinical safety data. Future comorbidity drug discovery must integrate organ-specific risk profiles using platforms like DrugBank's adverse effect atlas. Finally, this study was based on existing public databases and bioinformatics tools. Future research should incorporate advanced technologies, such as single-cell sequencing and CRISPR gene editing, to further elucidate the molecular mechanisms and specific pathways of crosstalk between AAA and periodontitis.

## 5. Conclusion

In conclusion, we identified IL1B, PTGS2, and SELL as key interacting genes in AAA and periodontitis through machine learning and various bioinformatics analysis methods, along with the significant roles of Tregs and neutrophils. IL1B and PTGS2 are both key regulators of inflammation, while SELL is involved in the migration and positioning of immune cells. This suggests that these factors could play significant roles in the shared immune mechanisms underlying both abdominal aortic aneurysm (AAA) and periodontitis. Our research provides fresh perspectives on the potential link between AAA and periodontitis, especially regarding immune cell movement and its impact on disease development. By investigating the expression patterns and mechanisms of these genes and immune cells in clinical samples, we aim to uncover new biomarkers and targeted therapeutic approaches for the early detection and treatment of both conditions. Future research should focus on confirming the exact roles of these genes and cells in AAA and periodontitis, particularly through the study of their expression, functions, and interactions in animal models and clinical samples. Clinical trials should explore the

potential of immunomodulatory and anti-inflammatory treatments based on these findings to alleviate symptoms in both AAA and periodontitis. Moreover, advanced techniques such as single-cell technologies and gene editing can offer deeper insights into the dynamic changes within immune cell subpopulations and their involvement in disease progression, potentially leading to more tailored therapeutic strategies for treating and preventing these two conditions. This study provides initial theoretical evidence for a link between AAA and periodontitis, but further experimental studies and clinical trials are necessary to confirm these results and assess their potential clinical implications. As such research advances, it is anticipated to offer fresh insights and approaches for the combined treatment and prevention of both conditions.

## Supporting information

**S1 Fig. Combine the periodontitis datasets (GSE10334 and GSE23586) into a single batch and perform PCA analysis.**
(TIF)

**S2 Fig. WGCNA was employed to identify modules associated with the synthetic periodontitis dataset.**
(TIF)

**S3 Fig. Based on the three key hub genes, drug target prediction was constructed, and Immunoscape analysis based on the three key hub genes of the AAA dataset.**
(TIF)

**S1 Appendix. This appendix documents the computational tools, R packages, and web resources utilized in the bioinformatic analysis of this study.** All resources are publicly available, with version numbers and access details specified to ensure reproducibility.
(DOCX)

## Acknowledgments

We sincerely thank the individuals and institutions who contributed to this study.

## Author contributions

**Data curation:** Chunjiang Liu, xiaoqi Tang, Zhaokun Sun.

**Formal analysis:** Yuan Wang, Chunjiang Liu.

**Methodology:** Zhifeng Wu, Fan Zhang, Liming Tang.

**Supervision:** Liming Tang.

**Validation:** Liming Tang.

**Visualization:** Zhifeng Wu, Fan Zhang, Liming Tang.

**Writing – original draft:** Zhifeng Wu, Fan Zhang, Yuan Wang.

**Writing – review & editing:** Zhifeng Wu, Liming Tang.

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
