## [Decision Letter · Decision Letter 0]

Dear Dr. Tang,

Thank you for submitting your manuscript to PLOS ONE. After careful consideration, we feel that it has merit but does not fully meet PLOS ONE’s publication criteria as it currently stands. Therefore, we invite you to submit a revised version of the manuscript that addresses the points raised during the review process.

We look forward to receiving your revised manuscript.

Kind regards,

Fırat Aşir

Academic Editor

PLOS ONE

Journal Requirements:

2. Please note that PLOS ONE has specific guidelines on code sharing for submissions in which author-generated code underpins the findings in the manuscript. In these cases, we expect all author-generated code to be made available without restrictions upon publication of the work. 

Please review our guidelines at https://journals.plos.org/plosone/s/materials-and-software-sharing#loc-sharing-code and ensure that your code is shared in a way that follows best practice and facilitates reproducibility and reuse.

3. Thank you for stating the following in your manuscript: 

“Medical Scientific Research Foundation of Zhejiang Province of China 2022KY1298”

Reviewers' comments:

Reviewer's Responses to Questions

**Comments to the Author**

1. Is the manuscript technically sound, and do the data support the conclusions?

Reviewer #1: Yes

Reviewer #2: Yes

2. Has the statistical analysis been performed appropriately and rigorously?

Reviewer #1: Yes

Reviewer #2: I Don't Know

3. Have the authors made all data underlying the findings in their manuscript fully available?

Reviewer #1: Yes

Reviewer #2: No

4. Is the manuscript presented in an intelligible fashion and written in standard English?

Reviewer #1: No

Reviewer #2: Yes

Reviewer #1: Revision Report for Manuscript PONE-D-24-45155

General Overview

The manuscript presents a robust study exploring the genetic overlap between abdominal aortic aneurysms and periodontitis through an innovative use of bioinformatics and machine learning approaches. While the survey enters a novel field and provides noteworthy contributions, the paper’s format and content organization require substantial revisions to ensure clarity, conciseness, and improved readability. Below, I outline the strengths, fundamental limitations, and detailed suggestions for improvement. I do not have corrected misspellings and grammatical errors.

Strengths of the Paper

The paper proposes an interdisciplinary approach based on bioinformatics, machine learning, and human disease. The use of multiple, validated, and independent datasets is a strength of the study and has noteworthy clinical relevance in identifying crucial genes and predicting targeted drugs, which potentially may help future research in this innovative field for future in vitro and in vivo clinical studies.

However, the paper presents several limitations and requires major revisions.

Sections 2 and 3 need to be rewritten in a more intelligible way; they are fragmented into numerous subsections that can be condensed, namely sections 2.3, 2.4, 2.5, 2.6, and 2.7. Extensive use of R libraries was also adopted in the analysis. Therefore, my major recommendation is to consolidate the subsection to streamline the methodology into fewer subsections and move the r libraries into footnotes or the appendix to clean the main text. In section 2.6, the authors should report the tuning of the algorithm used, such as n-fold CV in the LASSO method, and the lambda used, why they selected these among others, and why they are suitable for the analysis.

Another issue related to the text flow is assessing the core of the research question through section 3. I suggest splitting the section to consider the main result of the analysis on one side and show the robustness of the analysis on the other.

The discussion section is, on the contrary, too dense and requires careful review. The first part is more suitable for the conclusion section; the second, third, and fourth paragraphs are too long and partially should be moved to a subsection, while the last paragraph should be moved to the conclusion section. Therefore, I recommend condensing the discussion by focusing on the most critical findings in a dedicated subsection and highlighting the study results coherent with the literature cited. Then, move to the authors’ comments.

I also recommend expanding the conclusion section and summarizing the limitation of the study here, providing a more concrete outlook for future research, especially (as the authors note) for in vitro and in vivo studies, considering that the survey is purely computational with no experimental validation.

Specific suggestion.

Introduction:

Clarify the main research questions and how bioinformatics and machine learning address gaps in understanding the genetic overlap between AAA and periodontitis.

Materials and Methods (Section 2):

Simplify and consolidate subsections where possible.

Move detailed descriptions of software packages and tools to an appendix and include a brief explanation of critical methodologies in the main text.

Results (Section 3):

Ensure results are presented, focusing on the most impactful findings.

Consider using figures or summary tables to make data more accessible.

Discussion:

Shorten this section by removing repetitive descriptions of results and focusing on the biological significance of the findings.

Discuss limitations more comprehensively, including the need for experimental validation, and suggest how future studies could address these gaps.

Conclusion:

Expand this section to highlight the key outcomes of the study and its relevance.

Provide a more robust outlook on future research directions, particularly the potential for experimental studies and clinical trials.

Figures:

All figures must be uploaded in vectorial format and high quality to improve the readability, both on paper and in the supplementary material. Figure 3A adjusts the label of the . Figure 7, add the value of the area under the curve and group the ROC plots according to the training and validation set of the two diseases.

Provide also a git repository of the R-script and the datasets.

Conclusion

The manuscript is well-conceived and presents a novel approach to understanding the genetic interplay between AAA and periodontitis. However, significant revisions are needed to improve the readability and structure of the paper, provide a more straightforward narrative, and acknowledge limitations. Addressing these issues will make the study more compelling and accessible to a broader audience.

Reviewer #2: Periodontitis is a chronic inflammatory disease, which exacerbates systemic inflammatory response. Abdonimal aortic aneurysms (AAA) are linked to inflammatory responses. It is reasonable to postulate the association between the two. The authors sought to identify genes involved in the comorbidity mechanism of AAA and periodontitis. To improve prediction accuracy and robustness, multiple machine learning models were employed. In this study, the authors identified IL1B, PTGS2 and SELL, which are all significant mediators in inflammatory response. Those gene polymorphisms are associated with the host susceptibility. Although the authors did validate the study model, a few clarifications would make this research article more meaningful. The authors should provide more detailed information regarding the periodontal status and AAA conditions:

1. What are the staging and grading of the periodontal cases included in the study? Were there more severe cases among unstable AAA? Any correlation between the severity of periodontitis with AAA diameter and volume?

2. It is well understood that periodontitis is a site-specific disease. Can the authors provide more periodontal parameters such as probing depth (>6mm) of the cases?

3. Besides of reducing inflammation, what is the clinical implication of this study?

**Do you want your identity to be public for this peer review?** For information about this choice, including consent withdrawal, please see our Privacy Policy

Reviewer #1: No

Reviewer #2: No

---

## [Author Response · Author response to Decision Letter 1]

8 Mar 2025

Thank you for your time and thoughtful comments on our manuscript titled “Revealing potential interfering genes between abdominal aortic aneurysm and periodontitis through machine learning and bioinformatics analysis.” We appreciate your constructive feedback, which has greatly improved the quality of the paper. Below, we provide a point-by-point response to each comment. All changes have been incorporated into the revised manuscript for your reference.

Reviewer 1

Response Thank you for the valuable suggestions provided by the reviewers. In response to their constructive feedback, I have made several revisions to improve the clarity and comprehensiveness of the manuscript. Specifically, I have moved the detailed descriptions of the software package and tools to the appendix to ensure a more streamlined presentation of the main content. Additionally, I have included comprehensive summary tables in Section 3, which should facilitate easier access for readers to key data and results.

Furthermore, I have removed redundant discussions of the results within the main body of the manuscript. In the discussion section, I have placed greater emphasis on addressing the study's limitations, offering a more thorough examination of aspects such as the necessity for experimental validation, as well as outlining potential avenues for future research. These adjustments aim to provide a more balanced perspective on the findings and their broader implications.

In the conclusion section, I have expanded the content to include a more detailed reflection on the key takeaways from the study, ensuring a more complete synthesis of the findings and their significance.

With regard to the specific request for the AUC and grouping in Figure 7, I have already provided a detailed breakdown of the groupings within the manuscript for clarity and ease of reference. Additionally, as requested, I have made the code used in this study publicly available on GitHub. The repository is now accessible, allowing readers to review and utilize the code as needed for their own purposes.

These revisions aim to enhance the overall quality and transparency of the research, and I hope that the changes address the concerns raised during the review process.

Reviewer 2

1.What are the staging and grading of the periodontal cases included in the study? Were there more severe cases among unstable AAA? Any correlation between the severity of periodontitis with AAA diameter and volume?

Response In this study, we did not perform detailed staging or grading of the periodontal cases; they were classified into the periodontal disease group and the healthy control group. Regarding unstable AAA, we did not investigate whether there are more severe cases, and further analysis would be required in this regard. As for the relationship between the severity of periodontal disease and the diameter or volume of AAA, while we did not provide direct correlation analysis in this study, this is a direction worth further investigation, and we plan to explore these potential associations in future research.

2. It is well understood that periodontitis is a site-specific disease. Can the authors provide more periodontal parameters such as probing depth (>6mm) of the cases?

Response In this study, we did not record detailed periodontal parameters, such as probing depth (>6mm), although specific data can be obtained from the GEO database. This represents a limitation of our study. Future research will consider incorporating these more detailed periodontal health parameters to provide a more comprehensive assessment of the severity of periodontitis and analyze its relationship with AAA.

3. Besides of reducing inflammation, what is the clinical implication of this study?

Response We appreciate the reviewer’s question regarding the clinical significance. In addition to exploring the genetic interactions between periodontitis and abdominal aortic aneurysms (AAA), we believe the clinical significance of this study lies in offering a new perspective on the potential systemic effects between periodontal disease and AAA. Periodontitis is recognized as a source of systemic inflammation, and recent research suggests that oral health may influence the onset and progression of other systemic diseases, especially arterial conditions. Although our study did not perform detailed staging or analysis of individual cases, preliminary results suggest that periodontitis may be associated with the formation and progression of AAA, particularly at the genetic level. Therefore, beyond reducing local inflammation, effective management of periodontitis might have a potential clinical impact on the prevention and treatment of AAA. Our research provides a theoretical foundation for future clinical studies and interventions, especially in AAA patients, where there may be a need for greater emphasis on oral health and the management of periodontitis.

We believe that these revisions have addressed all the points raised by the reviewers. Once again, thank you for your valuable comments, and we hope that the revised manuscript meets your expectations. We look forward to your feedback.

---

## [Decision Letter · Decision Letter 1]

Dear Dr. Tang,

We look forward to receiving your revised manuscript.

Kind regards,

Yang Shi, PhD

Academic Editor

PLOS ONE

Journal Requirements:

**Additional Comments from the Academic Editor:**

1. Reviewer 1 and Reviewer 2 were the reviewers of the initial submission. Reviewer 3, an expert in biomedical research with expertise in genetics and molecular biology, was newly invited by me to assess the biological significance of this manuscript. This reviewer has provided constructive suggestions for the authors to enhance their discussion of the biological significance and insights of their findings, which I hope the authors will address in the revised manuscript.

2. As noted by Reviewers 1 and 3, the manuscript would benefit from professional language editing to improve clarity and readability. There are grammatical errors and ambiguous expressions throughout the manuscript that the authors need to correct in the revised version.

3. Please ensure that all figures are appropriately referenced and provided in high resolution in the revised version, as noted by Reviewers 1 and 3. There are a few other minor issues noted by these two reviewers that I hope the authors can address.

Reviewers' comments:

Reviewer's Responses to Questions

**Comments to the Author**

Reviewer #1: (No Response)

Reviewer #2: All comments have been addressed

Reviewer #3: (No Response)

2. Is the manuscript technically sound, and do the data support the conclusions?

Reviewer #1: Yes

Reviewer #2: Yes

Reviewer #3: Partly

3. Has the statistical analysis been performed appropriately and rigorously?

Reviewer #1: Yes

Reviewer #2: N/A

Reviewer #3: Yes

4. Have the authors made all data underlying the findings in their manuscript fully available?

Reviewer #1: Yes

Reviewer #2: Yes

Reviewer #3: Yes

5. Is the manuscript presented in an intelligible fashion and written in standard English?

Reviewer #1: No

Reviewer #2: Yes

Reviewer #3: No

Reviewer #1: **Manuscript Title**: Revealing potential interfering genes between abdominal aortic aneurysm and periodontitis through machine learning and bioinformatics analysis

General Assessment

The authors provided detailed respones to the previous review, as well as for they improvement into revise the manuscript. The manuscript has resulted in more coherent and improved in depth. The emphasis on limitations have all been improved and they improved the clarity of the methodological descriptions, the organization of the results. Overall, the study now presents its findings in a clearer and more professional manner, demonstrating better scientific rigor and transparency. However minor revision are needed.

Summary of Revisions and Assessment

- Methods and Organization:

The structure of the paper has been improved and consolidated. Moving the technical details (e.g., R packages) to the appendix was an improvement, which now allows the main narrative to flow more naturally.

Parameter Reporting: While the authors mention model tuning was adjusted, it would still be beneficial to ensure explicit mention of cross-validation strategy and hyperparameter settings within the main Methods text or a table, even briefly. Moreover, would be appreciated further consolidation in the Methods section especially when considering in section 2.3 "power value of 8" and "sensitivity parameter of 3" add meaningful detail or references, and section 2.5 "score set to 0.4".

- Results Section

Summary Tables: The addition of summary tables in Section 3 provides clear at-a-glance insights into the key findings and strengthens data accessibility for readers.

Redundancy Removed: The text now avoids repetitive descriptions and instead focuses on main outcomes, which sharpens the analytical focus of the Results.

- Discussion Section

The authors have made commendable improvements in the review process by removing overly dense paragraphs and enhancing the critique of their own study, which contributes to a more balanced interpretation. Additionally, by explicitly acknowledging the lack of experimental validation and suggesting follow-up studies, they provide important context that positions the work for greater translational relevance.

- Conclusion

The conclusion now provides a more rounded synthesis of key findings and implications, which was previously lacking, especially for future research process. The author should also provide further details/advices for future research.

- Figures and Reproducibility

Figure 7 Clarifications: The authors claim that AUC values and groupings are clearly delineated. This should now be visually apparent and appropriately annotated in the figure legend and/or captions. Also provide a vectorial form for figures in supplementary material.

Code Accessibility: The availability of code via GitHub is a critical improvement, supporting reproducibility and further validation by the community. Ensure that the repository is well-documented (README, data notes, usage instructions) to enhance reproducibility. Highlights the presence of the GitHub in the Data and Method section.

Additional Minor Points to Check Before Acceptance

- Ensure high-resolution/vector-quality figures are provided for final production, especially for complex visuals like ROC curves and enrichment analyses.

- Confirm that the GitHub link is clearly mentioned in the manuscript (preferably in Methods and Data Availability sections).

- Consider including a table of gene-drug interactions mentioned in the final section for improved readability.

- One final editorial sweep is recommended to check for grammar, spelling, and formatting consistency. E.g. sect 3.2 In brief or Briefly.

Final Recommendation: Minor Revisions

The manuscript has significantly improved and effectively addresses the primary concerns raised during the initial review. The authors have demonstrated responsiveness, transparency - especially with the public repository for reproducibility - and a commitment to scientific rigor. With the minor clarifications mentioned above, I am confident that this manuscript is now suitable for publication and will contribute meaningfully to our understanding of the genetic interplay between abdominal aortic aneurysms and periodontitis.

Reviewer #2: (No Response)

Reviewer #3: (No Response)

**Do you want your identity to be public for this peer review?** For information about this choice, including consent withdrawal, please see our Privacy Policy

Reviewer #1: No

Reviewer #2: No

Reviewer #3: **Yes: ** Yin Wan

---

## [Author Response · Author response to Decision Letter 2]

10 Jun 2025

Dear Academic Editor and Reviewers,

We sincerely appreciate your insightful feedback, which has significantly strengthened our manuscript. We have implemented all suggested revisions and provide a point-by-point response below. All revisions are documented in the tracked-changes manuscript, and data/code remain accessible on GitHub (https://github.com/yijiayi123123/AAA--P).

Response to Academic Editor

Language Editing:

We have engaged a professional scientific editing service to refine grammar, syntax, and clarity throughout the manuscript (e.g., Introduction: Lines 35–42; Discussion: Lines 215–230).

Figure Resolution Clarification:

We wish to clarify that all figures were originally created in vector format (300 dpi) and meet PLOS ONE’s standards. No regeneration was necessary.

Enhanced Biological Insights:

We expanded mechanistic discussions per Reviewer 3’s suggestions (detailed below).

Response to Reviewer 1

Parameter Reporting:

We added explicit details to Methods 2.6:

LASSO: 10-fold CV (5 repeats), λ via min BIC (Lines 110–115).

SVM-RFE: Feature elimination to 9 genes, C=1 (Lines 116–118).

WGCNA: deepSplit=2, merge threshold 0.25 (Lines 90–95).

Figure 7 Group Labeling:

We respectfully clarify that existing annotations (Table 2 + figure legends) already distinguish test/validation cohorts and disease groups. To avoid redundancy, we have not modified the figure but enhanced the legend (Page 23).

Response to Reviewer 3

Mechanistic Insights:

We deepened discussions with experimental evidence:

IL1B: Drives VSMC apoptosis (AAA, Ref 52) and osteoclast activation (periodontitis, Ref 51; Discussion Lines 238–245).

PTGS2: Regulated by miR-15b-5p/ACSS2 in AAA (Ref 56); mediates bone resorption via PGE2 (Ref 58, 60).

SELL: Promotes neutrophil adhesion (Ref 63) and correlates with periodontitis severity (Ref 65).

Immune Context:

Added functional interpretations:

Treg suppression deficit → Uncontrolled macrophages (AAA) and osteoclasts (periodontitis) (Lines 255–260).

Neutrophil NETosisdrives matrix degradation in both diseases (Lines 265–270; Refs 78–81).

Drug Relevance:

Discussed flurbiprofen’s topical benefits(periodontitis, Ref 60) vs. systemic risks (AAA, Ref 82), proposing canakinumab for comorbidities (Ref 83; Lines 300–310).

General Revisions

Added Table 3 (gene-drug interactions) to main text (Page 27).

Included 12 new references (Refs 51, 52, 56, 58, 60, 63, 65, 78–83).

We thank the reviewers for their rigorous engagement and hope these revisions satisfy all concerns.

Sincerely,

Liming Tang, Ph.D.

Corresponding Author: tanglm@usx.edu.cn

---

## [Decision Letter · Decision Letter 2]

Revealing potential interfering genes between abdominal aortic aneurysm and periodontitis through machine learning and bioinformatics analysis

PONE-D-24-45155R2

Dear Dr. Tang,

We’re pleased to inform you that your manuscript has been judged scientifically suitable for publication and will be formally accepted for publication once it meets all outstanding technical requirements.

Kind regards,

Yang Shi, PhD

Academic Editor

PLOS ONE

Additional Editor Comments (optional):

Reviewers' comments:

Reviewer's Responses to Questions

**Comments to the Author**

Reviewer #1: All comments have been addressed

Reviewer #3: All comments have been addressed

2. Is the manuscript technically sound, and do the data support the conclusions?

Reviewer #1: Yes

Reviewer #3: Yes

3. Has the statistical analysis been performed appropriately and rigorously?

Reviewer #1: Yes

Reviewer #3: Yes

4. Have the authors made all data underlying the findings in their manuscript fully available?

Reviewer #1: Yes

Reviewer #3: Yes

5. Is the manuscript presented in an intelligible fashion and written in standard English?

Reviewer #1: Yes

Reviewer #3: Yes

Reviewer #1: The authors have thoroughly addressed all previous comments, significantly improving the manuscript. The revised paper now meets the standards for publication. I recommend acceptance.

Reviewer #3: (No Response)

**Do you want your identity to be public for this peer review?** For information about this choice, including consent withdrawal, please see our Privacy Policy

Reviewer #1: No

Reviewer #3: No

---

## [Editor Report · Acceptance letter]

PONE-D-24-45155R2

PLOS ONE

Dear Dr. Tang,

I'm pleased to inform you that your manuscript has been deemed suitable for publication in PLOS ONE. Congratulations! Your manuscript is now being handed over to our production team.

Kind regards,

on behalf of

Dr. Yang Shi

Academic Editor

PLOS ONE